# Precise Epigenetic Analysis Using Targeted Bisulfite Genomic Sequencing Distinguishes FSHD1, FSHD2, and Healthy Subjects

**DOI:** 10.3390/diagnostics11081469

**Published:** 2021-08-13

**Authors:** Taylor Gould, Takako I. Jones, Peter L. Jones

**Affiliations:** Department of Pharmacology, University of Nevada, Reno School of Medicine, 1664 N. Virginia St., Reno, NV 89557, USA; tsantangelo@med.unr.edu

**Keywords:** muscular dystrophy, epigenetics, DUX4, D4Z4, FSHD, DNA methylation

## Abstract

The true prevalence of facioscapulohumeral muscular dystrophy (FSHD) is unknown due to difficulties with accurate clinical evaluation and the complexities of current genetic diagnostics. Interestingly, all forms of FSHD are linked to epigenetic changes in the chromosome 4q35 D4Z4 macrosatellite, suggesting that epigenetic analysis could provide an avenue for sequence-based FSHD diagnostics. However, studies assessing DNA methylation at the FSHD locus have produced conflicting results; thus, the utility of this technique as an FSHD diagnostic remains controversial. Here, we critically compared two protocols for epigenetic analysis of the FSHD region using bisulfite genomic sequencing: Jones et al., that contends to be individually diagnostic for FSHD1 and FSHD2, and Gaillard et al., that can identify some changes in DNA methylation levels between groups of clinically affected FSHD and healthy subjects, but is not individually diagnostic for any form of FSHD. We performed both sets of assays on the same genetically confirmed samples and showed that this discrepancy was due strictly to differences in amplicon specificity. We propose that the epigenetic status of the FSHD-associated D4Z4 arrays, when accurately assessed, is a diagnostic for genetic FSHD and can readily distinguish between healthy, FSHD1 and FSHD2. Thus, epigenetic diagnosis of FSHD, which can be performed on saliva DNA, will greatly increase accessibility to FSHD diagnostics for populations around the world.

## 1. Introduction

Facioscapulohumeral muscular dystrophy (FSHD) is typically an adult onset myopathy with variable penetrance that affects males and females of all ages [1,2]. There are two genetic classes of FSHD that are clinically indistinguishable [3]. The most common form, FSHD1, accounts for ~95% of cases and is caused by autosomal dominant heterologous deletions within the chromosome 4q35.2 D4Z4 macrosatellite array [4,5] that lead to a local epigenetic dysregulation of the contracted array [6,7]. The remainder of cases are classified as FSHD2, which has a digenic inheritance and is caused by mutations in genes that encode epigenetic repressors of the 4q35 and 10q26 D4Z4 arrays when combined with specific genetic requirements on 4q35 [8,9,10,11,12]. Therefore, the epigenetic disruption in FSHD2 is more widespread than in FSHD1 [13], affecting the D4Z4 arrays on all four alleles, while only the D4Z4 array on the contracted chromosome 4 is affected in FSHD1 [9,13]. Regardless, in all cases of FSHD, these mutations lead to the epigenetic disruption of the 4q35 D4Z4 array and increased expression of the pathogenic *DUX4* gene from the distal-most repeat unit (RU) within the array [14,15,16,17,18,19,20].

The pathogenic stable expression of the 4q35 *DUX4* gene highlights another unifying genetic feature of FSHD1 and FSHD2, the requirement for an FSHD permissive distal sequence [17,21,22]. Although hundreds of homologous D4Z4 repeats encoding similar *DUX* family sequences are scattered throughout the human genome, only the 4q35 and 10q26 D4Z4 arrays are epigenetically dysregulated in FSHD [23], and only gene expression from the 4q35 D4Z4 array is associated with FSHD [24]. Despite the entire *DUX4* protein coding sequence residing within each 4q35 and 10q26 D4Z4 repeat [14], these all lack the polyadenylation signal (PAS) required for the production of a stable mRNA. However, ~50% of human chromosome 4s contain a sequence immediately distal to the chromosome 4q35 D4Z4 array, termed 4qA [21], that contains a third exon with a PAS that, when spliced into the *DUX4* transcript, creates a mature *DUX4* mRNA leading to *DUX4* protein production [17]. These PAS-containing 4qA alleles are termed FSHD permissive chromosomes [17,21]. The remaining ~50% of chromosome 4s have an entirely different distal sequence, termed 4qB [21], and are FSHD nonpermissive since they do not contain a PAS and therefore cannot produce a stable *DUX4* transcript [17,21]. Contractions on 4qB alleles do not cause FSHD [22]. Further complicating matters, there are some specific sequence variations of 4qA that are not associated with FSHD. Thus, when performing FSHD genetic diagnostics, one must assay the specific D4Z4 array (chromosome 4q35), the specific D4Z4 distal sequence [25], and potentially the state of the FSHD2 genes (most commonly *SMCHD1*) [10].

As one might expect with such complicated genetics, FSHD is not amenable to a typical genetic analysis, such as using neuromuscular disease gene panels or whole exome sequencing, and instead requires specialized techniques specifically targeting FSHD [10,26,27,28,29,30]. To date, all the approved methods for FSHD1 genetic testing physically measure the size of the 4q35 D4Z4 arrays and thus require specially prepared high quality and high molecular weight genomic DNA (gDNA) and cannot be performed on saliva DNA or most biobanked DNAs [10,26,27,28,29,30]. These issues account for the fact that FSHD genetic testing is expensive, only performed at a few sites worldwide, and therefore inaccessible to many at-risk individuals around the world. More accessible FSHD genetic diagnostics are sorely needed.

Since the epigenetic dysregulation of 4q35 is common to both forms of FSHD and distinguishes FSHD from healthy subjects or those with non-FSHD myopathies [6,9,31], we developed a bisulfite sequencing (BSS)-based PCR (BS-PCR) approach for analyzing the methylation state of the key FSHD regions for use as a potential diagnostic (Figure 1, green, blue, and orange) [32]. While DNA hypomethylation has been used to help diagnose FSHD2 for a number of years [9,31,33], our method (Jones et al.) is currently the only one reported that can specifically identify both forms of FSHD using DNA methylation while concurrently distinguishing FSHD1 from FSHD2 [32]. Thus, we concluded that DNA methylation assayed by targeted BSS could be used as a molecular diagnostic for FSHD1 and FSHD2. Subsequently, a similar targeted BSS approach (Gaillard et al.) was published (Figure 1, pink) that identified gross differences in methylation levels between samples from healthy and clinically affected FSHD populations but with high overlap between the two groups, and did not find significant methylation differences between a cohort of genetically healthy (non-FSHD) individuals and asymptomatic subjects with genetic FSHD1 [34]. This result was in contrast to two other studies on the epigenetics of asymptomatic subjects that reported significantly higher D4Z4 methylation than in clinically affected FSHD1 subjects, but significantly lower methylation than in genetically healthy subjects [35,36]. In addition, although FSHD2 has been generally characterized by the FSHD field as distinctly different from FSHD1 and healthy subjects by hypomethylation of both the 4q35 and 10q26 D4Z4 arrays, shown by <30% DNA methylation using methylation sensitive restriction assay for the proximal *Fse*I site [10,13,37], Gaillard et al. reported no significant methylation differences between groups or individual cases of FSHD1 and FSHD2 while reporting strikingly high levels of DNA methylation (40–70%) in their FSHD2 samples [34]. The authors concluded that clinical FSHD is epigenetically distinct from healthy controls; however, DNA methylation does not distinguish all individuals with an FSHD D4Z4 contraction from those without and does not distinguish FSHD1 from FSHD2. Further studies continued to show significant differences between the epigenetic signatures obtained using the two protocols [34,36,38,39,40,41,42], leading some in the field to dismiss DNA methylation as not being diagnostic for, or even relevant for FSHD [43], with the discrepancies accounted for by different patient populations and potential technical concerns.

Recently, the genetic connection between FSHD2 and arhinia/Bosma arhinia microphthalmia syndrome (BAMS), caused by mutations in the *SMCHD1* gene [38,39], has led to a wider interest in the epigenetic analyses of the 4q35 locus using these two protocols. Thus, it is even more important and timely to clarify the discrepancies between the analyses. Here, we investigated the differing results and interpretations between these two techniques by performing a direct comparison using the original published protocols [32,34] on the same gDNA samples. We found that a lack of amplicon specificity for the BS-PCR primers used to amplify the 4q35 and 10q26 D4Z4 regions in the assay by Gaillard et al. [34] accounts for the different results between the two techniques. Importantly, we show that DNA methylation of the 4q35 D4Z4 arrays, when accurately and specifically assayed [32,36], is in fact diagnostic for FSHD and distinguishes FSHD1 from FSHD2. In addition, DNA methylation data obtained using oligonucleotide sequences (including more recent NGS-based analysis) from the Gaillard et al. study [34] should be interpreted with extreme caution [38,40,41,42].

## 2. Materials and Methods

### 2.1. Subjects

This study was approved by the University of Nevada, Reno Institutional Review Board (#1316095, approved on 9 October 2018). All participants that provided saliva samples (designated by PLJ in Table 1) signed informed consent. FSHD2 myoblasts (MB) and fibroblasts (FB) were obtained de-identified from Dr. Rabi Tawil, University of Rochester Medical Center [44]. All samples were obtained from participants that had undergone prior genetic testing for FSHD. Subjects C-01, C-02, C-03, C-04, C-05, C-06, C-07, C-08, and C-09 were clinically healthy and genetically confirmed as not having FSHD1 genetics (both chromosome 4q D4Z4 arrays longer >10 RUs or >48 kb EcoRI/BlnI fragment). Subjects F1-01, F1-02, F1-03, F1-04, F1-05, F1-06, F1-07, F1-08, and F1-09 were all clinically FSHD (based on prior examination) and genetically confirmed FSHD1 (<11 D4Z4 RUs or <38 kb EcoRI/BlnI fragment on a permissive 4qA chromosome). Subjects F2-01, F2-02, F2-03, F2-04, F2-05, and F2-06 were clinically FSHD (based on prior examination) and genetically confirmed as FSHD2 by the identification of a known FSHD2 mutation in the *SMCHD1* gene. Subject F2-01 was genetically diagnosed with both FSHD1 and FSHD2.

### 2.2. Sample Collection and DNA Preparation

Oragene DNA saliva collection kits (DNA Genotek, Ottawa, Canada) were used to collect saliva samples for gDNA isolation (Subjects C01~C05, F1-01~-05, F2-01 and F2-02). Fibroblasts (Subjects F2-03, F2-05, F2-06) and myoblasts (Subject F2-04) were cultured, as previously described [36], for gDNA isolation. The gDNA was isolated using the Wizard Genomic DNA Purification Kit (Promega, Madison, WI, USA) and prepIT•L2P (DNA Genotek).

### 2.3. DNA Methylation Analysis

Genomic DNA samples (1.5 µg) were bisulfite converted using the EpiTect Bisulfite Kit (Qiagen, Germantown, MD, USA) following the manufacturer’s protocol. All PCRs were performed using a BioRad C1000 Touch thermal cycler. The 4qA specific BSSA (300 ng of converted DNA), 4qAL specific BSSL (150 ng of converted DNA), and 4q/10q D4Z4 specific BSSX (150 ng of converted DNA) BS-PCRs were performed as described [32,36]. The BIS-3′, BIS-5′, and BIS-Mid BS-PCRs (150 ng of converted DNA for each) used published oligonucleotide primers (Appendix A, with corrected orientation) [34] and appropriate thermocycler conditions to amplify these D4Z4 regions. Briefly, the BIS-3′, BIS-5′, and BIS-Mid regions were amplified as follows: 94 °C for 2 min, 35 cycles of 94 °C for 15 s, 54 °C for 15 s, 72 °C for 25 s, followed by 72 °C for 10 min. All PCRs used GoTaq HS Polymerase (Promega) and all oligonucleotide primers used are listed in Appendix A. All PCR products were cloned into the pGEM-T Easy vector (Promega) and at least 15 independent colonies were sequenced for each region and analyzed using the bisulfite analysis website, BISMA (http://services.ibc.uni-stuttgart.de/BDPC/BISMA/, accessed on 30 June 2021) with default parameters [45]. Any sequences corresponding to 4A166 or 10A166, if present due to rare aberrant amplification, were readily identified by SNPs and eliminated from the analysis.

## 3. Results

Here, we addressed the conflicting published data regarding FSHD-relevant DNA methylation of the chromosome 4q35 and 10q26 D4Z4 macrosatellite repeat regions from two similar approaches [32,34] by directly comparing the results of the protocols performed on the same genomic DNA samples. The Jones et al. method analyzes two regions of the FSHD-associated D4Z4 array, one specific for the distal-most chromosome 4qA or 4qAL RU (BSSA or BSSL) and another that specifically amplifies all 4q and 10q D4Z4 RUs (BSSX) (Figure 1). Concerns raised with the Jones et al. protocol are: (1) the inclusion of a CpG in one of the primer sequences may bias the results with respect to methylation status and (2) the distal-most D4Z4 repeat may not be representative of the epigenetics of the whole D4Z4 array [32,40,43]. In contrast, the Gaillard et al. method analyzes three regions of the D4Z4 RU (BIS-5′, BIS-mid, and BIS-3′), all of which are present in all chromosome 4q and 10q D4Z4 RUs (Figure 1), and does not include any CpG in the primer design. However, one serious concern is the large number of missing predicted CpGs in the products assayed despite a >90% sequence identity cut-off for analysis, suggesting amplification of alternative D4Z4 loci [23,34]. The two protocols assay slightly different regions of D4Z4 and a direct comparison cannot be made for all of them (Figure 1); however, the BIS-3′ amplicon of the Gaillard et al. protocol is completely contained within the BSSA amplicon of the Jones et al. protocol (Figure 1, Figure 2, Figure 3 and Figure 4), thereby allowing at least one direct comparison of specificity and interpretation of results between the two assays. An additional complication of comparing the two approaches is the metric each uses for determining if a sample is hypomethylated or not; the Gaillard et al. approach uses the overall average methylation of all BS-PCR products analyzed in each assay, while the Jones et al. approach uses a quartile analysis for distinguishing between FSHD and healthy (BSSA or BSSL assay) and the average methylation for distinguishing FSHD1 from FSHD2 (BSSX assay). Therefore, for this comparative study, we used both sets of metrics on both sets of assays (Table 2 and Table 3).

Genomic DNA (gDNA) samples were obtained from genetically confirmed healthy, FSHD1, and FSHD2 subjects (Table 1) and analyzed using targeted BSS for DNA methylation by both the Jones et al. and Gaillard et al. BSS protocols [32,34]. Methylation analysis of the gDNAs from five healthy individuals (Figure 2, Appendix A) showed that both sets of assays similarly found all five samples to be hypermethylated > 35%, regardless of using average DNA methylation or quartile analysis (Table 2, Table 3, and Appendix A), as was expected for healthy controls and consistent with prior published results using the assays [9,23,32,34,36,46]. However, upon viewing the individual chromosome methylation map reads for the two assays, we found a disconcerting difference in the data obtained from the two approaches. The Jones et al. assay showed almost complete coverage (>98%) of all expected CpGs in both regions amplified based on the chromosome 4q35 D4Z4 sequence, as indicated by mostly blue and red squares (Figure 2, Appendix A). In contrast, two of the assays in the Gaillard et al. protocol, BIS-Mid and BIS-3′, produced methylation signatures that were missing many CpGs based on the reference D4Z4 sequences being amplified, as indicated by the numerous white squares. In fact, in the five control subjects analyzed, the BIS-Mid assay only identified 75.8%, 82.1%, 77.0%, 79.1% and 83.6% of the expected CpGs, while the BIS-3′ assay identified only 67.9%, 69.3%, 70.9%, 71.8% and 65.6% of the expected CpGs in the amplified regions from the chromosome 4q35 D4Z4 (Appendix A). This is despite the BISMA analysis software using the standard metrics for inclusion of >95% conversion rate and a 90% lower threshold for sequence identity [45], which indicates that the amplified sequences are highly similar to the expected sequences over their entire length but not with respect to the CpG sites. It should be noted that the Gaillard et al. study utilized the BiQ Analyzer software [47] for their methylation analysis, presumably under default parameters, instead of BISMA, although this should not change the output data. Regardless, since we also slightly altered the PCR conditions from the published method and used a different bisulfite conversion protocol for standardization [34], we reviewed the primary data in the original Gaillard et al. paper for a comparison. The figures presented show that their control methylation data and graphic methylation representations were almost identical to ours, showing hypermethylation from all three amplicons and many expected CpGs missing from the BIS-Mid and BIS-3’ assays despite the same 90% lower threshold sequence identity cutoff [34], validating our results using their primer sets. Regardless, despite these concerns, in these healthy subjects, all assays tested and analyzed by either method indicated hypermethylated D4Z4s, as expected [9,13,31,48]. From an FSHD diagnostics standpoint, each healthy individual would be correctly characterized by both assays (Table 2 and Table 3).

We similarly analyzed gDNA obtained from five FSHD1 subjects (Figure 3 and Appendix A) and six FSHD2 subjects (Figure 4 and Appendix A) using both assays. Again, the BSSA, BSSX, and BIS-5’assays identified nearly 100% of the expected CpGs in all five samples while the BIS-MID and BIS-3’ assays failed to identify ~25% of the expected CpGs (74.6% to 83.6% and 70.5% to 74.6%, respectively, Appendix A) for each subject, as indicated by white boxes in the graphic representations. However, for these FSHD samples, the reported methylation profiles produced significantly different results and diagnostic interpretations. The BSSA assay was designed to distinguish FSHD from healthy methylation based on <25% methylation for the first quartile (Q1) if an individual has two FSHD permissive 4A161 chromosomes or the second quartile (Q2) if an individual has one FSHD permissive 4A161 chromosome (Table 1). It should be noted that the 4A166 allele, while technically FSHD permissive due to the presence of the *DUX4* PAS in exon 3, is not associated with FSHD and is not amplified by the BSSA assay [32,36,49]. Thus, for subjects such as F1-03 and F2-06 that are 4A161/4A166, Q2 is used for the key methylation assessment. Overall, this quartile methylation metric revealed FSHD levels of DNA hypomethylation for all fourteen genetically confirmed FSHD subjects in this study (Table 2), consistent with prior reports [32,36]. The BSSX assay was designed to distinguish FSHD1 methylation from FSHD2 methylation using the average methylation for all 4q35 and 10q26 D4Z4 BS-PCR products analyzed. This assay showed the expected FSHD1 methylation signature (>35%) for all five FSHD1 subjects and the expected FSHD2 methylation signature (<30%) for all six FSHD2 subjects (Table 2, Figure 5 and Appendix A). Thus, the analysis using the Jones et al. assays was consistent with prior published data [32,36,39,50] and consistent with known FSHD epigenetics [9,10,13,31,46,48]. In contrast, the BIS-5′, BIS-Mid, and BIS-3′ assays found no differences in DNA methylation between the FSHD1 and healthy subjects (Table 3). Those assays fared slightly better with FSHD2. While the BIS-5′ assay found characteristic FSHD2 hypomethylation (<30%) in four out of the six FSHD2 samples, the BIS-Mid and BIS-3′ assays found no differences between FSHD2 and healthy subjects (Table 3 and Figure 5). Similarly, these results are consistent with the published data using these assays [34,38,40,42].

The discrepancy between the techniques with respect to FSHD2 methylation is particularly concerning as DNA hypomethylation (<30% on average) of all four of the relevant chromosome 4q35 and 10q26 D4Z4 arrays, as determined by MSRE assay, has long been considered diagnostic for FSHD2 [10,25]. Similarly, the DR1 BSS assay independently validated 4q35 and 10q26 D4Z4 hypomethylation as a signature of FSHD2 [33]. Therefore, we plotted the FSHD2 data obtained from all five assays for comparison (Figure 5). Conceptually, the BSSX, BIS-5′, BIS-Mid, and BIS-3′ each amplify from all four relevant D4Z4 arrays (Figure 1). Consistent with previous independent data, the BSSX assay reports <30% average methylation for all six FSHD2 samples (Table 2 and Figure 5). However, the average methylation as well as the vast majority of BSS reads from the BIS-Mid and BIS-3′ assays never reach near to the predicted FSHD2 levels of hypomethylation. The BIS-5′ assay, which has the best concordance of analyzed CpGs to expected CpGs of the three Gaillard et al. amplicons (Appendix A), reports median FSHD2 levels of methylation for four of the six FSHD2 samples (Table 3 and Figure 5). This further supports that the Jones et al. amplicons are specifically assaying the relevant FSHD regions and the Gaillard et al. amplicons, and the BIS-mid and BIS-3’ in particular, are likely not specific to the FSHD region.

Overall, the presented data supports that the reported discrepancies for FSHD1 and FSHD2 methylation between the two BSS methods are not due to differences between the individuals being assayed or to epigenetic and genetic differences within the 4q35 D4Z4 RUs [40], but are instead due to differences in the assays themselves, especially with regard to specificity. It should be noted that there are several other D4Z4 arrays in the human genome, as discussed below, that have highly similar sequences to the 4q and 10q D4Z4s, but are genetically distinct and not epigenetically dysregulated in FSHD1 or FSHD2 [23].

As shown here and in previous studies [32,36,50], the BSSA assay very accurately identifies FSHD-specific hypomethylation in both forms of FSHD, but not controls. However, a concern was raised that the BSSA results may not be a true representation of the potentially varied methylation states of the chromosome 4 D4Z4s in all cells and instead could be selectively amplifying hypomethylated chromosomes due to the design of the oligonucleotide primers used [43]. The basis of this concern is that some of the oligonucleotide primers used in the Jones et al. assays encompass sequences that contain CpGs. Since CpGs can exist as either methylated or unmethylated, and the two states would give a different DNA sequence in the bisulfite-converted DNA (CpG when methylated or UpG when unmethylated), a primer designed against one or the other would preferentially amplify that state of DNA. Thus, CpGs are typically avoided in BSS primer design when an unbiased result is desired. However, this bias can be avoided, as we have done. In the BSSA amplicon, the two nested reverse primers are in a region that contains a single CpG, which was necessary to achieve the desired specificity for the distal 4qA D4Z4 RU. Similarly, the reverse primer for the BSSX amplicon contains two CpGs. Since specificity for the correct D4Z4 repeat array was the most important factor in primer design, these CpG base pairs were included, but only after taking into consideration the two possible states. As shown in Appendix A, primers BSSA-3742R, BSSA-3626R, and BSSX-1036R contain an “R” designation at the CpGs in question, which corresponds to either a G or A at that base randomly included in the primer synthesis. Since there is only one “R” base pair per BSSA primer sequence, statistically 50% of the primers will be 100% complementary to the methylated state (a G to base pair with a C) and 50% of the primers will be 100% complementary to the unmethylated state (an A to base pair with the U) and they will be in equal abundance in the BS-PCR. The difference in melting temperatures between the “A” and “G” primers is negligible. However, to determine experimentally if there is any preference for a methylation state in this assay, four additional subjects (for a total of five) with genetically confirmed FSHD1, a single contracted 4qA allele (hypomethylated), and a second noncontracted 4qA allele (hypermethylated) and three additional healthy subjects (for a total of four) with two noncontracted 4qA alleles were tested using the BSSA assay. If there was a preference for one state or the other, we would expect a shift in the results away from 50%. As expected, we found two roughly equivalent pools of chromosomes with respect to methylation state in FSHD1 (Table 2, Appendix A and Appendix A), with no apparent preference for the methylation state of the amplified chromosome, as indicated by the ready amplification of chromosomes with between 0 to 80% methylation in FSHD1 or 20 to 78% in the controls. 

Overall, this direct comparison of two commonly used BSS methods for FSHD indicates that the Jones et al. methodology produces an accurate assessment of the DNA methylation state of the FSHD-associated chromosome 4q35 D4Z4 array that is consistent with prior epigenetic analyses of FSHD using other methods [6,7,9,13,31,46], and accurately distinguishes FSHD from healthy, as well as distinguishing FSHD1 from FSHD2. In contrast, the Gaillard et al. methodology does not distinguish FSHD from healthy, does not accurately assess FSHD1 methylation, and only occasionally correctly identifies an FSHD2 methylation state with one of their three assays. Therefore, published data produced from the Gaillard et al. assays, including the recent NGS version based on the same primer sequences, should be interpreted with extreme caution [34,40,42,43,51].

## 4. Discussion

The question has been raised, “Does DNA methylation matter in FSHD?” [43] There are two key requirements for addressing this important question: (1) the DNA methylation status of the FSHD locus must be accurately and specifically analyzed, and (2) the subjects analyzed must have a correct clinical diagnosis. Here, we further validated our BSS-based epigenetic diagnostic protocol for FSHD [32] and directly addressed a controversy in the field regarding the utility of BSS assays for accurately assessing the epigenetic status of the chromosome 4q35 and 10q26 D4Z4 repeat arrays. A prior attempt to reconcile the differences between two commonly used assays found the same reported discrepancy and thus came to the conclusion that the distal-most D4Z4 RU, which is specifically assayed in the Jones et al. BSSA method, must be epigenetically distinct from the rest of the contracted (FSHD1) 4q35 or dysregulated 4q35 and 10q26 (FSHD2) D4Z4 RUs [40]. However, their analysis only showed summaries of the methylation data that could not be evaluated for sequence integrity. We believe the simpler and more likely explanation is that the Gaillard et al. methodology results in amplification of divergent D4Z4s that are irrelevant to FSHD.

There are many similar D4Z4 repeat arrays in the human genome, yet only the chromosome 4q35 and 10q26 D4Z4 arrays, which have >98% sequence identity between their respective D4Z4 RUs and are the only D4Z4 RUs to encode an intact *DUX4* open reading frame (ORF) [5,28,52,53], are relevant to FSHD genetics and epigenetics [23]. Thus, when investigating FSHD epigenetics, it is extremely important to assay only these specific D4Z4 arrays and RUs. To accomplish this difficult feat, the original technique to assess the epigenetic state of the D4Z4 regions, methyl-sensitive restriction enzyme digestions (MSRE) using *Fse*I digestion of the proximal 4q and 10q D4Z4 RU (Figure 1), utilized Southern blotting probed with the p13E-11 DNA sequence that is specific for the region centromeric to these two D4Z4 arrays [5,6,9,10,13,31,46]. These studies showed that only the contracted 4q35 D4Z4 array was hypomethylated in FSHD1 while all four 4q35 and 10q26 D4Z4 arrays were hypomethylated in FSHD2. Subsequently, multiple independent BSS assays (Hartweck et al., Jones et al., and Calandra et al.) were used to confirm the allele-specific FSHD1 hypomethylation profile [7,32,36] and the broader FSHD2 hypomethylation profile [7,32,33]. Only the Gaillard et al. BSS analysis has continually failed to produce these distinct hypomethylation profiles, and the graphic representations of the data, when shown, strongly support that the nonspecific amplification of divergent D4Z4 RUs is to blame. 

The D4Z4 RUs from chromosomes 4q and 10q are nearly identical (>98%) across their entire 3.3kb DNA sequences [5,28,52,53], including most of the CpGs, with only a couple of nucleotide variants [23], and this is supported by the near 100% coverage of expected CpGs shown in the graphic representation of the data from the Jones et al. BS-PCR amplicons (Figure 2, Figure 3 and Figure 4 and Appendix A, red and blue boxes). In addition, long-read sequencing through a 13 RU 4qA D4Z4 BAC shows a nearly identical sequence for each D4Z4 RU [28]. Conversely, the D4Z4 arrays on chromosomes 3, 13, 14, 15, 21, 22, and the Y chromosome, which are not epigenetically dysregulated in either form of FSHD, have 30–60 nucleotide variants per RU just in the ~500 bp region corresponding to their putative *DUX4* ORFs, and 70% of those variants are C/G to A/T transitions, thus eliminating potential methylation sites [23]. BS-PCR amplification and analysis from these divergent D4Z4 RUs would not be eliminated by a 90% lower threshold cut-off for sequence identity and would allow for the inclusion of these hypermethylated BSS reads in all samples, which would skew the analysis towards more reported methylation. A tell-tale sign that this is happening would be the absence of multiple predicted CpGs (shown as white boxes in the graphic representations of methylation data) and suspiciously high methylation in FSHD2 samples. As shown in Figure 4 and Appendix A, the BIS-Mid and BIS-3′ assays likely amplify these divergent and perpetually hypermethylated D4Z4 RUs. Interestingly, in the graphic representations of the FSHD2 data for the BIS amplicons (Figure 4 and Appendix A), which are ordered by highest (top) to lowest (bottom) methylation, the few sequences with 100% CpG coverage (no white boxes) are also those with the lowest percent methylation (bottom) and are more consistent with FSHD2 levels (<30%) of methylation. In addition, considering that the region amplified by the BIS-3’ assay, with numerous missing CpGs in the analysis, is completely contained within the BSSA assay region (Figure 1, Figure 2, Figure 3 and Figure 4, Appendix A), which identified >98% of expected CpGs, it is clear that the BIS-3′ assay is not specific for the 4q D4Z4, while the BSSA assay is specific. Thus, we conclude that the data obtained from these two amplicons is nonspecific for the 4q and 10q D4Z4 arrays and thus not relevant for FSHD. The BIS-5′ assay appears more specific than the other Gaillard et al. assays; however, since it analyzes a region that can only be used to assess FSHD2, the results of this assay have no bearing on FSHD1 epigenetics.

The above data supports that, when properly analyzed, the BSS analysis of the FSHD locus is a diagnostic for genetic FSHD1 and FSHD2 [32], which would represent a significant breakthrough in the field with respect to global accessibility. Traditional FSHD diagnostics, using pulsed-field gel electrophoresis and Southern blotting, and the new single molecule fluorescent techniques physically measure the sizes of the 4q35 D4Z4 arrays [25,26,27,29,30] and long-read diagnostic DNA sequencing assays being developed for sequencing through an entire FSHD1-contracted D4Z4 array [28,54] all require high quality HMW gDNA and cannot be performed on the gDNA isolated from saliva samples or on biobanked gDNA, greatly limiting their applicability. In contrast, epigenetic analysis using BSS is PCR-based and can be performed on gDNA isolated from any source, including saliva, which can be collected through the mail, making this type of genetic testing highly accessible. However, due to the referenced conflicting published BSS data for FSHD between Jones et al. (2014) [32] and Gaillard et al. (2014) [34], there has been a disagreement in the field as to which, if any, BSS-based analysis accurately represents the epigenetic status of the chromosome 4q35 disease locus. We have now clarified the FSHD BSS landscape and conclude that, in theory, BSS using the Jones et al. method could be used for accurate molecular diagnosis of genetic FSHD1 and FSHD2. However, larger controlled cohorts need to be analyzed and reported before this technique is put into clinically relevant practice.

## 5. Conclusions

We directly addressed two conflicting reports on the utility of BSS analysis for FSHD diagnostics and interpretations for FSHD and arhinia epigenetic studies. We determined that the discrepancies are due to differences in specificity of the amplicons for the FSHD locus. We conclude that the original Jones et al. protocol (2014) is highly specific for the FSHD-associated chromosome 4q35 D4Z4 array and is diagnostic for distinguishing FSHD from healthy subjects and FSHD1 from FSHD2 subjects. In contrast, the Gaillard et al. protocol (2014) does not accurately report the methylation state of the FSHD locus and instead nonspecifically amplifies related but unlinked D4Z4 repeats that are not epigenetically dysregulated in FSHD, thereby skewing the data towards reporting an inaccurate, more methylated state [23]. Thus, DNA methylation data obtained for FSHD and arhinia studies based on the Gaillard et al. BS-PCR amplicons should be interpreted with caution, if not revisited.

## Figures and Tables

**Figure 1 diagnostics-11-01469-f001:**
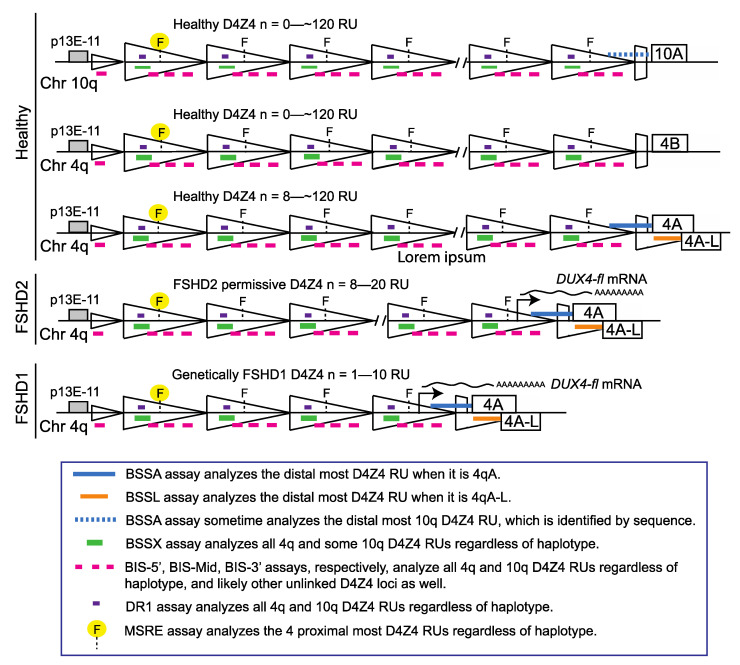
Schematic of FSHD-associated chromosome 4q D4Z4 regions analyzed by BSS (modified from [32]). The genomic regions assayed by the three most commonly used methods for FSHD diagnostics are shown. The methylation sensitive restriction enzyme (MSRE) digestion and Southern blotting method assays the methylation state of the most proximal *Fse*I restriction enzyme site (yellow highlight) on both alleles for the 4q and 10q chromosomes when probed with p13E-11 [25,30]. The DR1 BSS assay identifies FSHD2-specific DNA hypomethylation [33]. The BSS assays used by Gaillard et al. separately amplify three different regions of the D4Z4 (BIS-5′, BIS-mid, BIS-3′; pink) that are also present on both alleles of the 4q and 10q chromosomes [34]. In contrast, the BSS assays used by Jones et al. amplify two products that are specific for the distal-most chromosome 4q, one for 4qA alleles (BSSA, blue), and the other for 4qA-L alleles (BSSL, orange), and one product upstream of the *DUX4* ORF that is present on all four alleles (BSSX, green) [32]. The genomic region amplified by the BIS-3′ assay is completely within the BSSA amplified region and the DR1 region is completely within the BSSX region.

**Figure 2 diagnostics-11-01469-f002:**
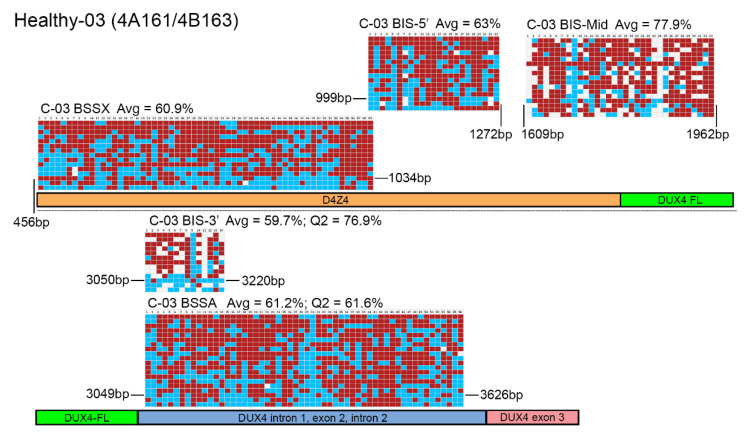
BSS analysis comparison of healthy epigenetic signatures. BS-converted DNA from a genetically confirmed healthy subject (C-03) was analyzed using the BSSA, BSSX, BIS-5′, BIS-Mid, and BIS-3′ assays. Each BS graphic representation is shown approximately where it aligns with each part of the 4q35 D4Z4 array including the D4Z4 (orange), *DUX4-fl* ORF (green), *DUX4* Intron 1, Exon 2, Intron 2 (purple), and Exon 3 (pink), with exact location indicated by base pairs (bp) downstream from the *Kpn*I site of the distal D4Z4 RU. Blue boxes indicate unmethylated CpGs, red boxes indicate methylated CpGs, and white boxes indicate no CpG where one is expected. Average methylation percentages are denoted “Avg” and have values of 60.9% for BSSX, 63% for BIS-5′, 77.9% for BIS-Mid, 59.7% for BIS-3′, and 61.2% for BSSA. This control subject has a haplotype of 4A161/4B163. Therefore, the diagnostic methylation percentages for the second quartile (Q2) are included for BSSA (61.6%) and BIS-3′ (76.9%) for comparison.

**Figure 3 diagnostics-11-01469-f003:**
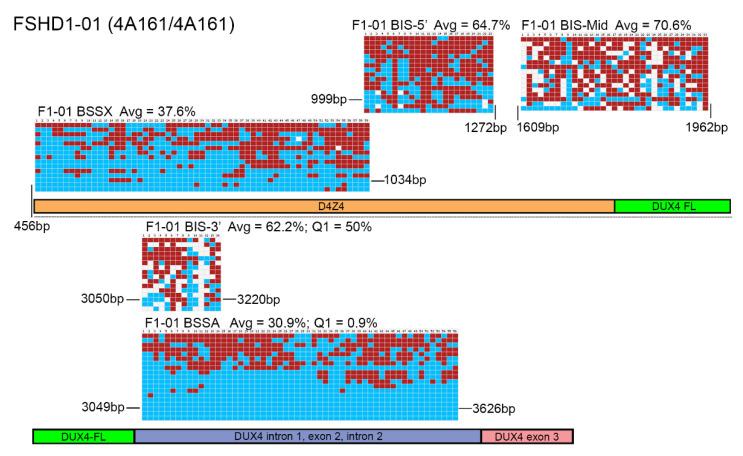
BSS analysis comparison of FSHD1 epigenetic signatures. BS-converted DNA from a genetically confirmed FSHD1 subject (F1-01) was analyzed using the BSSA, BSSX, BIS-5′, BIS-Mid, and BIS-3′ assays. Each BS graphic representation is shown approximately where it aligns with each part of the 4q35 D4Z4 array including the D4Z4 (orange), *DUX4-fl* ORF (green), *DUX4* Intron 1, Exon 2, Intron 2 (purple), and Exon 3 (pink), with exact location indicated by base pairs (bp) downstream from the *Kpn*I site of the distal D4Z4 RU. Blue boxes indicate unmethylated CpGs, red boxes indicate methylated CpGs, and white boxes indicate no CpG where one is expected. Average methylation percentages are denoted “Avg” and have values of 37.6% for BSSX, 64.7% for BIS-5′, 70.6% for BIS-Mid, 62.2% for BIS-3′, and 30.9% for BSSA. This FSHD1 subject has a haplotype of 4A161/4A161. Therefore, the diagnostic methylation percentages for the first quartile (Q1) are included for BSSA (0.9%) and BIS-3′ (50%) for comparison.

**Figure 4 diagnostics-11-01469-f004:**
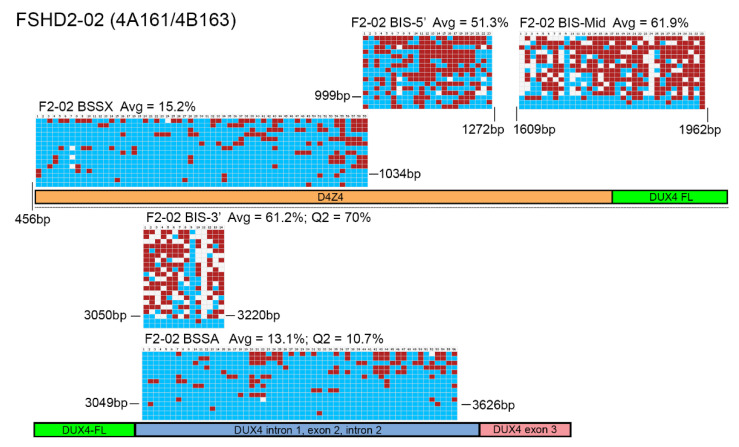
BSS analysis comparison of FSHD2 epigenetic signatures. BS-converted DNA from a genetically confirmed FSHD2 subject (F2-02) was analyzed using BSSA, BSSX, BIS-5′, BIS-Mid, and BIS-3′ assays. Each BS graphic representation is shown approximately where it aligns with each part of the 4q35 D4Z4 array including the D4Z4 (orange), *DUX4-fl* ORF (green), *DUX4* Intron 1, Exon 2, Intron 2 (purple), and Exon 3 (pink), with exact location indicated by base pairs (bp) downstream from the *Kpn*I site of the distal D4Z4 RU. Blue boxes indicate unmethylated CpGs, red boxes indicate methylated CpGs, and white boxes indicate no CpG where one is expected. Average methylation percentages are denoted “Avg” and have values of 15.2% for BSSX, 51.3% for BIS-5′, 61.9% for BIS-Mid, 61.2% for BIS-3′, and 13.1% for BSSA. This FSHD2 subject has a haplotype of 4A161/4B163. Therefore, the diagnostic methylation percentages of the second quartile (Q2) are included for BSSA (10.7%) and BIS-3′ (70%) for comparison.

**Figure 5 diagnostics-11-01469-f005:**
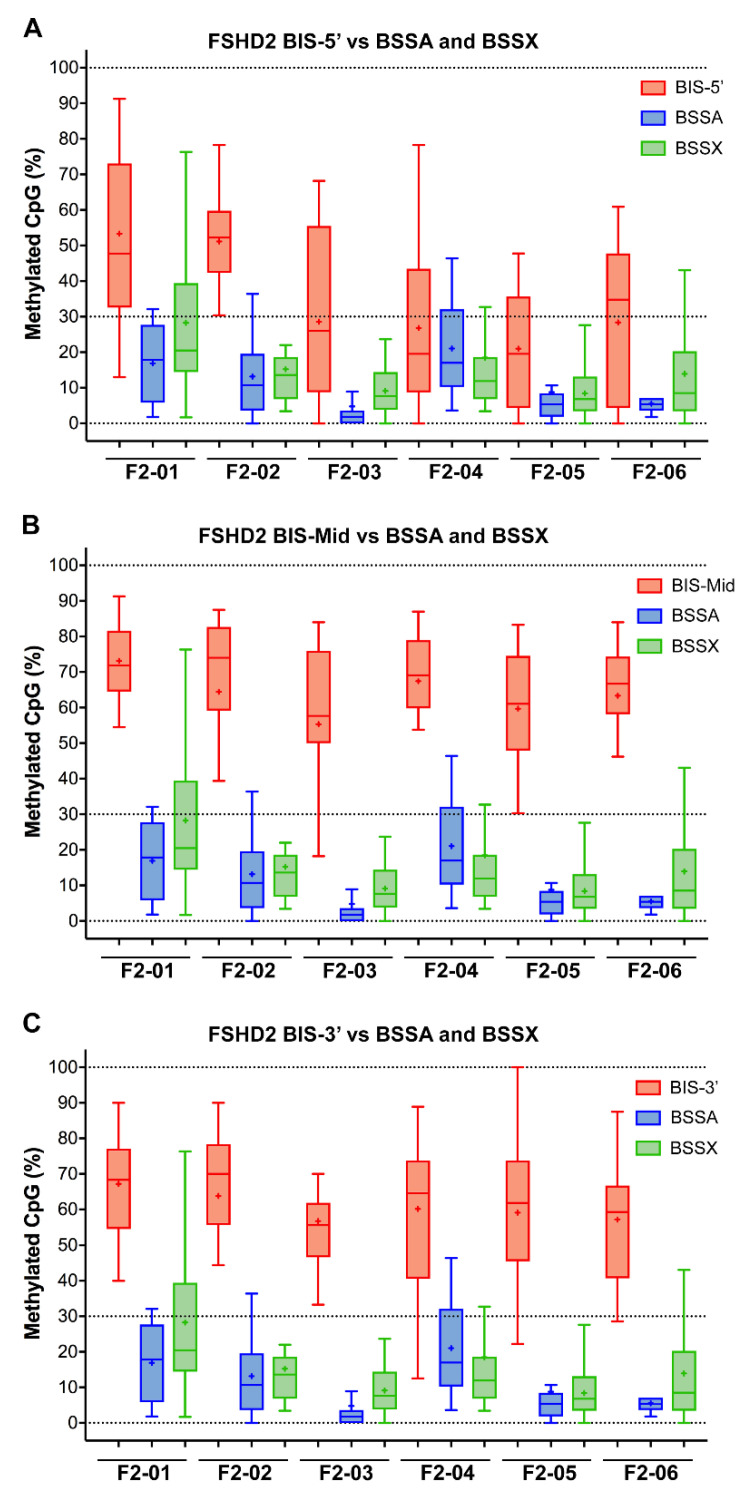
Box plots showing the percentage of CpGs analyzed for each single BSS read for each individual FSHD subject comparing the three BIS assays to the BSSA and BSSX assays. (**A**) The BIS-5′ assay (red) is compared with the BSSA (blue) and BSSX (green) assays. (**B**) The BIS-Mid assay (red) is compared with the BSSA (blue) and BSSX (green) assays. (**C**) The BIS-3′ assay (red) is compared with the BSSA (blue) and BSSX (green) assays. The box extends from 25 to 75 percentiles with whiskers showing the smallest and largest value. The bar in the box is the median and the + is the mean for each sample.

**Table 1 diagnostics-11-01469-t001:** Genetic characterization of the samples used in the study.

Subject	Sample ID	Genetic Diagnosis	Haplotype	DRA (E/B)	D4Z4 RUs	*SMCHD1*
C-01	PLJ-10171	Healthy	4A161/4A161L	>48 kb	ND	ND
C-02	PLJ-10186	Healthy	4A161/4A161L	>48 kb	ND	ND
C-03	PLJ-10167	Healthy	4A161/4B163	>48 kb	ND	ND
C-04	PLJ-10212	Healthy	4A161/4B162	>48 kb	ND	ND
C-05	PLJ-20084	Healthy	4A161/4A161	>48 kb	ND	ND
C-06	PLJ-10111	Healthy	4A161/4A161	>48 kb	ND	ND
C-07	PLJ-10255	Healthy	4A161/4A161	>48 kb	ND	ND
C-08	PLJ-20034	Healthy	4A161/4A161	>48 kb	ND	ND
F1-01	PLJ-20043	FSHD1	4A161/4A161	12 kb	3	ND
F1-02	PLJ-10123	FSHD1	4A161/4A161	15 kb	4	ND
F1-03	PLJ-10194	FSHD1	4A161/4A166	35 kb	10	ND
F1-04	PLJ-10218	FSHD1	4A161/4B163	25 kb	7	ND
F1-05	PLJ-10163	FSHD1	4A161/4B163	23 kb	6	ND
F1-06	PLJ-10278	FSHD1	4A161/4A161	35 kb	10	ND
F1-07	PLJ-20052	FSHD1	4A161/4A161	12 kb	3	ND
F1-08	PLJ-20030	FSHD1	4A161/4A161	15 kb	4	ND
F2-01	PLJ-10185	FSHD1 + 2	4A161/4B163	35 kb	10	c.2914−5A>G
F2-02	PLJ-20068	FSHD2	4A161/4B163	>48 kb	ND	c.2146+1G>A
F2-03	18MB054	FSHD2	4A161/4A161	46 kb	13	1.2Mb deletion
F2-04	18MB052	FSHD2	4A161/4B163	42 kb	12	c.3444T>A
F2-05	19FB042	FSHD2	4A161/4B168	43 kb	12	c.610A>G
F2-06	19FB047	FSHD2	4A161/4A166	46 kb	13	1.2 Mb deletion

DRA (E/B) = D4Z4 reduced allele using the EcoRI/BlnI size in kilobases (kb); D4Z4 RUs = repeat units rounded to the nearest integer (± 1 RU); ND = not determined.

**Table 2 diagnostics-11-01469-t002:** Results of epigenetic analysis using the Jones et al. method.

Subject	BSSA Avg	BSSA Q1	BSSA Q2	BSSA Q3	BSSX Avg	BSSX Q1	BSSX Q2	BSSX Q3	Epigenetic Diagnosis
C-01	58	51.8	54.5	65.2	48	24.55	45.3	68.95	Healthy
C-02	54.4	48.2	54.35	62.5	62	46.6	71.2	76.3	Healthy
C-03	61.2	56.75	61.6	67.9	60.9	52.55	66.1	72.05	Healthy
C-04	61.8	52.7	61.6	70.5	53.9	38.15	55.9	67.8	Healthy
C-05	52.8	42.9	56.25	64.3	56.8	34.2	64.4	78	Healthy
C-06	64.6	59.8	69.1	73.2	55.3	46.65	52.5	65.25	Healthy
C-07	58.2	52.25	55.4	65.15	48.1	31.6	43.25	61.85	Healthy
C-08	52.1	42.9	51.8	60.7	37.5	20.03	35.05	44.95	Healthy
F1-01	30.9	0.9	28.6	58.95	37.6	22	33.9	58.95	FSHD1
F1-02	48.1	7.1	39.3	67	50.6	35.6	48.3	68.65	FSHD1
F1-03	27.3	23.2	25.9	33.9	53.6	10.2	71.2	86.4	FSHD1
F1-04	21.4	11.7	20.5	26.15	37.21	20.3	37.3	60.7	FSHD1
F1-05	15.4	10.7	16.1	21.4	61.6	57.8	68.25	73.3	FSHD1
F1-06	35.2	8.9	28.55	62.5	51.3	33.1	55.2	71.2	FSHD1
F1-07	31.9	1.8	27.95	68.75	64.2	49.15	69.5	78	FSHD1
F1-08	32.9	8	14.3	60.35	63.8	61	67.8	79.7	FSHD1
F2-01	16.9	6.25	17.85	26.8	27.6	15.3	20.5	36.2	FSHD2
F2-02	13.1	4.55	10.7	19.6	15.2	8.55	13.6	17.75	FSHD2
F2-03	4.8	0.9	1.8	3.6	9.1	4.25	7.65	13.6	FSHD2
F2-04	21.1	10.7	17	32.1	12.2	5.1	7.75	16.9	FSHD2
F2-05	6.4	1.85	5.4	7.1	8.4	3.4	6.8	12.75	FSHD2
F2-06	5.5	3.6	5.4	7.1	13	3.4	8.5	20.3	FSHD2

The Jones et al. BSSA analysis utilizes the relevant quartile (yellow boxes), which corresponds to Q1 for those with two 4A161 alleles and Q2 for those with one 4A161 allele, to determine the methylation status. Conversely, the Gaillard et al. analysis utilizes the average methylation (orange column) to determine the methylation status of a sample. Both analyses would use average methylation for BSSX assessment (green column). The epigenetic diagnosis is based on the Jones et al. method metrics. ND = not determined.

**Table 3 diagnostics-11-01469-t003:** Results of epigenetic analysis using the Gaillard et al. method.

Subjects	BSS 3′ Avg	BSS 3′ Q1	BSS 3′ Q2	BSS 3′ Q3	BSS Mid Avg	BSS Mid Q1	BSS Mid Q2	BSS Mid Q3	BSS 5′ Avg	BSS 5′ Q1	BSS 5′ Q2	BSS 5’ Q3	Epigenetic Diagnosis		
C-01	59.1	54.5	63.05	75	74.6	69.7	75	81.4	69.5	54.35	76.1	87	Healthy		
C-02	60.6	37.5	70	73.85	74.2	63.8	77.3	82.4	67.8	56.5	67.4	80.45	Healthy		
C-03	59.7	32.15	76.9	81.65	77.9	72.35	77.3	84.6	63	50	64.4	69.8	Healthy		
C-04	71.3	66.7	75	78.9	72.1	66.7	71.4	76	73	62.05	76.1	82.6	Healthy		
C-05	62.6	50	63.05	73.85	78	67.95	81.5	85.9	69.5	54.45	73.9	78.3	Healthy		
F1-01	62.2	50	68.35	79.3	70.6	63.35	75	79.75	64.7	52.3	65.2	82.65	Healthy		
F1-02	65.9	45.85	71.4	80	73.7	66.45	74.55	84.4	73.4	56.5	80.45	91.3	Healthy		
F1-03	69.9	64.6	71.35	80	75.8	66.05	77.1	85.95	73.1	63.05	73.3	89.15	Healthy		
F1-04	61.7	47.75	66.7	73.85	69.9	64.1	70.45	76.8	75.2	65.2	80.45	91.3	Healthy		
F1-05	56.3	47.2	52.8	65.15	77.6	66.05	76	88.95	58.7	39	56.5	78.3	Healthy		
F2-01	67.5	54.5	68.35	76.4	73	64.5	71.9	81.65	53	34.75	47.8	72.25	Healthy		
F2-02	61.2	55.6	70	78.4	61.9	59.2	74	82.05	51.3	42.5	52.2	58.7	Healthy		
F2-03	60.4	52.25	61.25	70	53.5	50	57.7	75.95	28.1	8.75	26.1	55.5	H	H	F
F2-04	61.4	43.75	64.6	72.5	67	60.5	69.1	78.75	26.4	8.7	19.55	43.5	H	H	F
F2-05	57.7	45.5	61.8	72.5	59.2	48	61.05	74.1	20.8	4.3	19.55	34.8	H	H	F
F2-06	58.3	41.45	59.3	66.7	62.5	58.3	66.7	73.9	27.8	10.85	34.8	47.7	H	H	F

Gaillard et al. analysis utilizes the average methylation (orange columns) to determine the methylation status of a sample. The Jones et al. analysis utilizes the relevant quartile (yellow boxes), which corresponds to Q1 for those with two 4A161 alleles and Q2 for those with one 4A161 allele, to determine the methylation status. The epigenetic diagnosis is based on the Gaillard et al. method metrics. H = healthy, F = FSHD.

## Data Availability

All data is reported in this manuscript and supplementary information.

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
