# Peer review of "Precise Epigenetic Analysis Using Targeted Bisulfite Genomic Sequencing Distinguishes FSHD1, FSHD2, and Healthy Subjects"

_diagnostics, 2021, doi:10.3390/diagnostics11081469_

Round 1

Reviewer 1 Report

The search for DNA methylation differences ad diagnostic methods to couple and improve genetic testing is a timely and attractive issue. In the present study the authors compared two published methylation protocols observing that their method performs better than another one in distinguish FSHD1, FSHD2 and healthy controls than the other one. Despite that results are extremely interesting, the manuscript is biased by the fact that autors are the developers and proponents of one of the two protocols. I would recommend to send samples to an independent lab. for protocols testing and validation of the results. 

Author Response

The search for DNA methylation differences and diagnostic methods to couple and improve genetic testing is a timely and attractive issue. In the present study the authors compared two published methylation protocols observing that their method performs better than another one in distinguish FSHD1, FSHD2 and healthy controls than the other one. Despite that the results are extremely interesting, the manuscript is biased by the fact that authors are the developers and proponents of one of the two protocols. I would recommend sending samples to an independent lab for protocols testing and validation of the results.

Response: Conceptually, we agree that when two groups have conflicting data, performance of the study by a third independent party, as suggested here, is useful for resolving the dispute. However, this request is not practical, considering that we were given ten days to respond to these reviews and there are significant MTA and funding issues that would need to be addressed. More importantly, we feel that this approach is not relevant to this situation, since the dispute is not about differing data, but about differing interpretations of what is essentially the same data.

The results that we obtained using the Gaillard et al protocol are virtually identical to the results obtained by the authors of that study. We are not claiming a different result, but a different interpretation. We provide additional data that clearly explains and validates this. Clearly, their data is contaminated with that from a disease-irrelevant region due to non-specific amplification, and since this is not recognized and the sequences are not removed, their conclusions are incorrect. Importantly, the difference between our groups is not due to different samples, as has been suggested, but the fact that their assay is not specific for the disease-relevant region.

In fact, the Magdinier lab (of Gaillard et al.) performed a similar experiment in a study published in 2019 (Roche et al., Neurology Genetics 5:e372) where they used both techniques on their samples, albeit without providing the key graphic information. Using our assay, they confirmed the methylation differences that we find between FSHD and control subjects, despite using a different metric (mean
methylation) and failing to take haplotype into account. Thus, they obtained data consistent with ours, yet still failed to realize that their assay is not specific. They instead concluded that the last D4Z4 repeat unit must be epigenetically different from all others in the array. Again, we disagree with the conclusion.

We believe that a third party would undoubtedly produce the same data presented here using each of the published techniques, as both are highly reproducible. If the graphic data were presented as ours is (showing the presence or absence of all expected CpGs in sequenced clones), we anticipate that they would agree with our interpretation and explanation, as the second reviewer has, now having been provided with the detailed evidence presented here. Timely publication of this study will allow others to make their own evaluations of the data and draw their own conclusions. Anyone who is interested could then certainly conduct their own third-party evaluation, having all data in hand.

By presenting all data from both protocols and the same samples, we have shown it is consistent with prior published data using these protocols, and propose a difference in specificity as the clear reason for the differing interpretations.

We trust that this addresses the concern.

Reviewer 2 Report

The manuscript by the Jones group accurately compares two previously describes BSS methods for the analysis of methylation in FSHD. They find out that the BS-primers  used in the Gaillard BSS method most probably amplify D4Z4-like sequences from other loci (than 4q35 and 10q26) that have a methylation status independent from FSHD. As a consequence, the Gaillard BSS method is not capable of detecting relevant methylation changes between FSHD and control and between FSHD1 and FSHD2. Jones and colleagues presented very valuable information because the authors of the Gaillard method recently published a manuscript were the question the relevance of methylation in FSHD diagnosis. This manuscript shows that the Gaillard methylation analysis fails and that methylation does matter in FSHD. This paper also supports the methylation studies previously presented by other groups using different methylation analysis methods. Therefore this is an important paper to be published.

I only have a few minor comments

The authors clearly show that the Jones methylation assay can discriminate FSHD from controls and FSHD1 from FSHD2. It is very promising that the method does not require HMW DNA, but instead can be done on saliva DNA or poor quality liquid DNA.

  • Is the methylation level comparable between saliva DNA and blood DNA?
  • Based on the presence of one or two 4A161 alleles they use BSSA Q1 or Q2 to determine the methylation. Most permissive 4qA alleles have the haplotype 4A161, but there are some exceptions (hybrid 4A166H or the rare haplotypes 4A159, 4A163 and 4A168). Luckily, these permissive haplotypes can all be recognized by the BSSA assay, but some of these alleles might be hard to identify using (SSLP-)PCR only. It also seems important to know prior to the analysis if the 4A161 allele has an S or L distal sequence. Possibly, the DNA samples that they use have been analyzed in detail prior to their methylation analysis. How do you do the methylation analysis and interpretation of the results on DNA samples that have not been accurately genotyped before?

Introduction page 2, line 58.

The authors shortly mention sequence variations of 4qA not associated with FSHD. They probably refer to the 4A166 allele.

I miss the reference here to the 4A166 paper. But more importantly, the authors forget to mention that they BSSA assay that they developed is does not amplify the 4A166 allele (and not 10qA and all B-type alleles) due to a SNP in primer BSSA-3626R. This is a great feature and makes the BSSA methylation assay more specific for FSHD-permissive alleles. Probably, this specificity has been mentioned in previous papers by your group, but here this might help to understand the results and analysis for individuals F1-03 and F2-06 (in table 2 and Figure S4). Both individuals carry two 4qA alleles, but show an epigenetic signature of only one allele in Figure S4 (F1-03) and in table 2 the BSSA Q2 was used to determine the methylation status, because the homologues 4qA is a 4A166 allele.

In figure 1, the authors mention a repeat size range for 4qA (4A and 4A-L) in control individuals  between 11-120 units. Based on other studies, this should be between 8-120 units as about 2% of the control individuals carry a repeat size between 8-10 units.

In table 1, please also give the D4Z4 repeat array size in units (or use units instead of DRA[E/B])

And if complete genotyping has been done, please also show the D4Z4 repeat size of the homologous 4qA allele. This information is important for researchers that are interested in the BSSA methylation performance for >10 units D4Z4 repeats

FSHD1 patient F1-06 carries a DRA(E/B) of 38 kb, which is 10 or 11 units. This is quite unusual for FSHD1 and more common for FSHD2. Unfortunately the BSSX analysis to identify FSHD2 was not performed on this individual. On the other hand, the high BSSA Q3 level (62.5%) found for this individual already seem to exclude FSHD2. What is your opinion on this?

Figure 2-4, For clarity, please mark the location of the different BSS amplicons within or on top of the orange-green-blue-pink marked D4Z4 array.

Page 8, line 269. The authors mention 11 genetically confirmed FSHD subjects. I don’t understand this number. Maybe 14 (FSHD1+FSHD2)?

Page 8, line 271, BSSX assay using the average methylation on all 4q35 D4Z4 BS-PCR products. Maybe, I misunderstood. Do you mean, average methylation for all chromosome 4q35 and 10q26 D4Z4 arrays?

Table S3

Primer BSSA-1428F, was previously named BSSA-1438F.

Figure S2 legend 4A161L allele is hypomethylated. Is this correct? Probably hypermethylated

Author Response

The manuscript by the Jones group accurately compares two previously describes BSS methods for the analysis of methylation in FSHD. They find out that the BS-primers used in the Gaillard BSS method most probably amplify D4Z4-like sequences from other loci (than 4q35 and 10q26) that have a methylation status independent from FSHD. As a consequence, the Gaillard BSS method is not capable of detecting relevant methylation changes between FSHD and control and between FSHD1 and FSHD2.
Jones and colleagues presented very valuable information because the authors of the Gaillard method recently published a manuscript were the question the relevance of methylation in FSHD diagnosis. This manuscript shows that the Gaillard methylation analysis fails and that methylation does matter in FSHD.
This paper also supports the methylation studies previously presented by other groups using different methylation analysis methods. Therefore this is an important paper to be published.

I only have a few minor comments

The authors clearly show that the Jones methylation assay can discriminate FSHD from controls and FSHD1 from FSHD2. It is very promising that the method does not require HMW DNA, but instead can be done on saliva DNA or poor quality liquid DNA.

• Is the methylation level comparable between saliva DNA and blood DNA?
Response: Yes, this is documented in our 2014 Clinical Epigenetics publication.

• Based on the presence of one or two 4A161 alleles they use BSSA Q1 or Q2 to determine the methylation. Most permissive 4qA alleles have the haplotype 4A161, but there are some exceptions (hybrid 4A166H or the rare haplotypes 4A159, 4A163 and 4A168). Luckily, these permissive haplotypes can all be recognized by the BSSA assay, but some of these alleles might be hard to identify using (SSLP-)PCR only. It also seems important to know prior to the analysis if the 4A161 allele has an S or L distal sequence. Possibly, the DNA samples that they use have been analyzed in detail prior to their methylation analysis. How do you do the methylation analysis and interpretation of the results on DNA samples that have not been accurately genotyped before?
Response: We currently completely haplotype all samples. Regardless, our current protocol is to perform the BSSA assay on every sample regardless of haplotype just in case there is a recombination event or something that might be otherwise misleading. We then follow up with the BSSL or BSS166 based on results from 4A/B PCR, SSLP, and if a BSSA product was obtained.

Introduction page 2, line 58.
The authors shortly mention sequence variations of 4qA not associated with FSHD. They probably refer to the 4A166 allele.
I miss the reference here to the 4A166 paper. But more importantly, the authors forget to mention that they BSSA assay that they developed does not amplify the 4A166 allele (and not 10qA and all B-type alleles) due to a SNP in primer BSSA-3626R. This is a great feature and makes the BSSA methylation assay more specific for FSHD-permissive alleles. Probably, this specificity has been mentioned in previous papers by your group, but here this might help to understand the results and analysis for individuals F1-03 and F2-06 (in table 2 and Figure S4). Both individuals carry two 4qA alleles, but show an epigenetic signature of only one allele in Figure S4 (F1-03) and in table 2 the BSSA Q2 was used to determine the methylation status, because the homologues 4qA is a 4A166 allele.
Response: We added in the phrasing “It should be noted that the 4A166 allele, while technically FSHD permissive due to the presence of the DUX4 PAS in exon 3, is not associated with FSHD and is not amplified by the BSSA assay. Thus, for subjects such as F1-03 and F2-06 that are 4A161/4A166, Q2 is used for the key methylation assessment.” and reference our 2014 Clinical Epigenetics paper and the 2020 Preston paper.

In figure 1, the authors mention a repeat size range for 4qA (4A and 4A-L) in control individuals between 11-120 units. Based on other studies, this should be between 8-120 units as about 2% of the control individuals carry a repeat size between 8-10 units.
Response: Genetically FSHD1 is still 1-10RUs but we understand that clinically most 8-10RUs are healthy in the absence of modifiers. We have made the change to the figure.

In table 1, please also give the D4Z4 repeat array size in units (or use units instead of DRA[E/B])
Response: We included both in the new version of Table 1.

And if complete genotyping has been done, please also show the D4Z4 repeat size of the homologous 4qA allele. This information is important for researchers that are interested in the BSSA methylation performance for >10 units D4Z4 repeats
Response: We agree this would be helpful, however, we do not have this information. Most tests were commercial and only indicate >48kb for the other allele.

FSHD1 patient F1-06 carries a DRA(E/B) of 38 kb, which is 10 or 11 units. This is quite unusual for FSHD1 and more common for FSHD2. Unfortunately, the BSSX analysis to identify FSHD2 was not performed on this individual. On the other hand, the high BSSA Q3 level (62.5%) found for this individual already seem to exclude FSHD2. What is your opinion on this?
Response: First, we rechecked the genetic reports and the 38kb was the Eco fragment not E/B so they are actually 35kb E/B. Regardless, we performed the BSSX analysis and added it to the Table 2.
They are clearly not FSHD2. However, this individual is clearly clinically affected but mild and not until late in life. In our experience, we have seen this now several times with 9-10RUs being clinically affected but not FSHD2. It is surprising but highlights that there is still a lot we do not know about FSHD pathogenic mechanisms.

Figure 2-4, For clarity, please mark the location of the different BSS amplicons within or on top of the orange-green-blue-pink marked D4Z4 array.
Response: We added the base pairs to the diagrams.

Page 8, line 269. The authors mention 11 genetically confirmed FSHD subjects. I don’t understand this number. Maybe 14 (FSHD1+FSHD2)?
Response: This refered to the 11 FSHD subjects analyzed using both sets of assays. An additional three FSHD1 4A161/4A161 subjects were later included to address preference for a methylated or unmethylated allele. We added some clarity to the sentence and report 14 subjects.

Page 8, line 271, BSSX assay using the average methylation on all 4q35 D4Z4 BS-PCR products. Maybe, I misunderstood. Do you mean, average methylation for all chromosome 4q35 and 10q26 D4Z4 arrays?
Response: yes, we have corrected it.

Table S3
Primer BSSA-1428F, was previously named BSSA-1438F.
Response: BSSA-1438 is correct and has been changed in the table S3

Figure S2 legend 4A161L allele is hypomethylated. Is this correct? Probably hypermethylated
Response: Yes, that was another typo and fixed to hypermethylated

Round 2

Reviewer 2 Report

No further comments, I am pleased with the revision